# Designing mRNA- and Peptide-Based Vaccine Construct against Emerging Multidrug-Resistant *Citrobacter freundii*: A Computational-Based Subtractive Proteomics Approach

**DOI:** 10.3390/medicina58101356

**Published:** 2022-09-27

**Authors:** Muhammad Naveed, Jawad-ul Hassan, Muneeb Ahmad, Nida Naeem, Muhammad Saad Mughal, Ali A. Rabaan, Mohammed Aljeldah, Basim R. Al Shammari, Mohammed Alissa, Amal A. Sabour, Rana A. Alaeq, Maha A. Alshiekheid, Safaa A. Turkistani, Abdirahman Hussein Elmi, Naveed Ahmed

**Affiliations:** 1Department of Biotechnology, Faculty of Science and Technology, University of Central Punjab, Lahore 54590, Pakistan; 2Department of Medical Education, Rawalpindi Medical University, Rawalpindi 46000, Pakistan; 3Molecular Diagnostic Laboratory, Johns Hopkins Aramco Healthcare, Dhahran 31311, Saudi Arabia; 4College of Medicine, Alfaisal University, Riyadh 11533, Saudi Arabia; 5Department of Public Health and Nutrition, The University of Haripur, Haripur 22610, Pakistan; 6Department of Clinical Laboratory Sciences, College of Applied Medical Sciences, University of Hafr Al Batin, Hafr Al Batin 39831, Saudi Arabia; 7Department of Medical Laboratory Sciences, College of Applied Medical Sciences, Prince Sattam bin Abdulaziz University, Al-Kharj 11942, Saudi Arabia; 8Department of Botany and Microbiology, College of Science, King Saud University, Riyadh 11451, Saudi Arabia; 9Department of Medical Laboratories Technology, Faculty of Applied Medical Science, Taibah University, Al Madinah Al Munawarh 42353, Saudi Arabia; 10Department of Medical Laboratory, Fakeeh College for Medical Science, Jeddah 21134, Saudi Arabia; 11Department of Medical Microbiology and Parasitology, School of Medical Sciences, Universiti Sains Malaysia, Kubang Kerian 16150, Malaysia

**Keywords:** antibiotic resistance, AMR, health issues, global hazards, bioinformatics

## Abstract

*Background and Objectives:* *Citrobacter freundii (C. freundii)* is an emerging and opportunistic Gram-negative bacteria of the human gastrointestinal tract associated with nosocomial and severe respiratory tract infections. It has also been associated with pneumonia, bloodstream, and urinary tract infections. Intrinsic and adaptive virulence characteristics of C. *freundii* have become a significant source of diarrheal infections and food poisoning among immune-compromised patients and newborns. Impulsive usage of antibiotics and these adaptive virulence characteristics has modulated the *C. freundii* into multidrug-resistant (MDR) bacteria. Conventional approaches are futile against MDR *C. freundii. Materials and Methods:* The current study exploits the modern computational-based vaccine design approach to treat infections related to MDR *C. freundii.* A whole proteome of *C. freundii* (strain: CWH001) was retrieved to screen pathogenic and nonhomologous proteins. Six proteins were shortlisted for the selection of putative epitopes for vaccine construct. Highly antigenic, nonallergen, and nontoxic eleven B-cell, HTL, and TCL epitopes were selected for mRNA- and peptide-based multi-epitope vaccine construct. Secondary and tertiary structures of the multi-epitope vaccine (MEVC) were designed, refined, and validated. *Results:* Evaluation of population coverage of MHC-I and MHC-II alleles were 72% and 90%, respectively. Docking MEVC with TLR-3 receptor with the binding affinity of 21.46 (kcal/mol) occurred through the mmGBSA process. Further validations include codon optimization with an enhanced CAI value of 0.95 and GC content of about 51%. Immune stimulation and molecular dynamic simulation ensure the antibody production upon antigen interaction with the host and stability of the MEVC construct, respectively. *Conclusions*: These interpretations propose a new strategy to combat MDR *C. freundii.* Further, in vivo and in vitro trials of this vaccine will be valuable in combating MDR pathogens.

## 1. Introduction

*Citrobacter* spp. are emerging and opportunistic dwellers of the human gastrointestinal tract, inhabitants of animals’ intestinal tracts, wastewater, and soil. *Citrobacter freundii* is a Gram-negative, facultatively anaerobic, nonsporulating, and rod-shaped bacillus bacterium that belongs to the family *Enterobacteriaceae* [1]. *Citrobacter freundii* is the most infectious species of *Citrobacter* spp. It participates in an array of nosocomial contagious diseases, including bloodstream infections (BSIs), respiratory tract infections (RTIs), biliary tract infections (BTI), and urinary tract infections (UTIs) [2]. It has also been involved in meningitis, endocardium, brain abscess, gastroenteritis, and pneumonia [3]. Certain *freundii* have also been reported to have intrinsic and adaptive virulence characteristics that cause diarrheal infections and food poisoning in the gastrointestinal tract of human beings, generally found in neonates and immunosuppressed patients [4].

Imprudent applications of antibiotics invoke several resistance mechanisms, prompting multidrug resistance (MDR) among various bacteria, especially Gram-negative bacteria [5,6]. Multidrug-resistant *Citrobacter freundii* has acquired resistance against multiple antibiotics, including extended-spectrum β-lactamase (ESBL), carbapenemase, and cephalosporins, and there are also strains of quinolone-resistant *Citrobacter freundii*. It has become a significant challenge for healthcare departments as most cases are associated with hospital-acquired infections, perpetuating hospitalization and immortality in patients relying on antibiotics [7].

Multidrug resistance of this opportunistic bacterium reduces the effectiveness of antibiotics so that novel antibiotics face resistance opposition strains of bacteria, consequently complicating the treatment of infected individuals. Immunotherapeutic and novel therapeutic approaches are considered to mitigate the emerging resistance mechanism in bacteria as there is no proper antibiotic to treat this infection-causing bacteria [8]. Formerly, vaccines were produced from attenuated, weakened, and dead pathogens, providing imperishable protection, but curtailing the vaccines’ swift and mass production of immunoinformatics-approach-based vaccine construct could be helpful in this regard as it involves specific antigens to the immune system to induce an adaptive immune response against bacterial infection, abating the time and cost of vaccine construction [9]. The computational-based peptide as well as mRNA vaccines have been designed against several viruses, such as COVID-19, and bacteria too [10,11,12].

The current study aims to design a novel peptide- and mRNA-based vaccine by evaluating *Citrobacter freundii*-related proteins assessed through proteome subtraction. Multiple computational-based bioinformatics tools were applied to predict PTO predicter T lymphocytes (HTL) and cytotoxic T lymphocytes (CTL) epitopes from the subtractive proteome. Putative epitopes were shortlisted based on outer membrane, pathogenicity, antigenicity, allergenicity, and immunogenicity. Highly antigen, nonallergen, and nontoxic peptide- and mRNA-based vaccines were constructed. Furthermore, the secondary and tertiary structures of the vaccine were predicted, codon-optimized, and validated through immune simulation and molecular dynamic simulation. These interpretations will assist researchers in developing precise and effective peptide- and mRNA-based vaccines against multidrug-resistant *Citrobacter freundii.*

## 2. Materials and Methods

### 2.1. Proteome Retrieval

The Universal Protein Resource (UniProt) was accessed for the subtraction of *Citrobacter freundii* (strain: CWH001) proteome through UniProt ID: UP000229029 (https://www.uniprot.org/) (accessed on 15 January 2022) [13] and chromosome region (Genome Accession: PEHH01000000) coding 5119 proteins [14]. The flowchart of the current study is shown in Figure 1.

### 2.2. Subcellular Localization and Host Homology

CELLO (http://cello.life.nctu.edu.tw/cello2go/) (accessed on 15 January 2022), a web server for the subcellular localization of proteins, was applied to screen pathogenic proteins based on their molecular localization function [15]. Pathogenic proteins homologous to the human host were eliminated using the BLASTp algorithm, keeping e-value: 10-5 as default parameters, and nonhomologous proteins were selected for further proceedings.

### 2.3. Protein Clustering, Antigenicity, and Allergenicity

The CD-HIT suite (weizhongli-lab.org) a database for clustering protein sequences, was applied for screening nonconcurrent and contrasting protein sequences with default parameters of 0.8 and 80% as cut-off value and identity percentage, respectively [16]. Vaxijen was performed with a threshold value of (>5) for the prediction of antigenicity (http://www.ddg-pharmfac.net/vaxijen/VaxiJen/VaxiJen.html) (accessed on 17 January 2022) [17], and allergenicity was predicted using Aller TOPv 2.0 (https://www.ddg-pharmfac.net/AllerTOP/) (accessed on 17 January 2022) [18]. Pathogenic proteins for the selection of putative vaccine targets were selected for epitope prediction having high antigenicity and nonallergen characteristics.

### 2.4. Identification of Trans-Membrane Alpha-Helices

To evaluate whether screened proteins were embedded in the cell membrane, we ran the TMHMM v.2.0 web server (DTU Health Tech, Kgs. Lyngby, Denmark) [19]. The hidden Markov model was applied to protein structural topology.

### 2.5. Prediction of Immune Cell Epitopes and Population Coverage Analysis

Putative B-cell or antibody epitopes, helper T-cell lymphocytes (HTL), and cytotoxic T-cell lymphocytes (CTL) were predicted utilizing different online servers. Immune Epitope Database (IEDB), an online server, was performed under the Bepipred linear epitope prediction 2.0 method to predict linear B-cell epitopes (http://tools.iedb.org/bcell/) (accessed on 19 January 2022) [20]. Immune Epitope Database MHC also predicted cytotoxic T-cell lymphocytes (CTL) epitopes-I (http://tools.iedb.org/main/tcell/) (accessed on 19 January 2022), where ANN 4.0 method was performed, sorting predicted epitopes by IC50 value, for the prediction of CTL epitopes [21]. Further, Immune Epitope Database MHC-II predicted the helper T-cell lymphocytes by performing the NN-align method (http://tools.iedb.org/main/tcell/) (accessed on 20 January 2022) [22]. Population coverage analysis was performed using the population coverage server of IEDB (http://tools.iedb.org/population/) (accessed on 20 January 2022) to recognize the interaction between MHC-I and MHC-II and their respective T-cell lymphocyte epitopes [23].

### 2.6. Designing of mRNA-Based Vaccine Construct 

The best putative epitopes were selected to design an mRNA-based vaccine against *Citrobacter freundii.* They predicted antigenic and nonallergen 11 cytotoxic T-cell lymphocytes, helper T-cell lymphocytes, and 11 B-cell or antibody epitopes from six protein sequences. Putative epitopes for vaccine design were linked through linkers (EAAAK, AAY, PMGLP, and GGGGS) [24] along with Kozak sequence, five G Cap, signal peptide, five untranslated regions (UTR), and addition of stop codon before three untranslated areas (UTR) with poly-A tail [25].

### 2.7. Designing of Peptide-Based Vaccine Construct 

The construction of a peptide-based vaccine was similar to the mRNA vaccine construct, except that adjuvants such as β-defensin epitopes were joined together through EAAK, AAY, GPGPG, and KK linker, and the β-defensin adjuvant. B-cell, cytotoxic T-cell, and helper T-cell epitopes from six shortlisted proteins were linked through EAAK, AAY, GPGPG, and KK linkers and adjuvants [24].

### 2.8. Physiochemical Properties, Antigenicity, Allergenicity, Toxicity, and Solubility Analysis of Vaccine Construct

Physiochemical properties of MEVC were predicted by applying the Expassy Protpram tool (https://web.expasy.org/protparam/) (accessed on 22 January 2022) [26]. Antigenicity and allergenicity were indicated by using Vaxijen (http://www.ddg-pharmfac.net/vaxijen/VaxiJen/VaxiJen.html, accessed on 22 January 2022) and Aller TOPv 2.0 (https://www.ddg-pharmfac.net/AllerTOP/, accessed on 22 January 2022). Toxicity analysis was performed using the ToxinPred webserver (http://crdd.osdd.net/raghava/toxinpred/, accessed on 24 January 2022) [27], and for the solubility analysis, the SOLUPROT tool (https://loschmidt.chemi.muni.cz/soluprot/ accessed on 24 January 2022) [28] was utilized for the solubility analysis of the vaccine construct.

### 2.9. Structures Prediction and Validation of Vaccine Construct

PSIPRED, an online server (http://bioinf.cs.ucl.ac.uk/psipred/, accessed on 28 January 2022), was run to predict the secondary structure of the vaccine construct under default parameters [29]. RoseTTAFold was performed to predict the precise 3D design of the vaccine construct. Moreover, the vaccine construct was refined by utilizing Galaxyweb (https://galaxy.seoklab.org/, accessed on 1 February 2022) [30], PROCHECK [31], and ERRAT [32] servers (https://saves.mbi.ucla.edu/, accessed on 1 February 2022), which were applied for the stereochemistry analysis using the Ramachandran plot [33], and atomic interaction of MEVC. Further validations were performed by utilizing proSA-web (https://prospro SAvices.came.sbg.ac.at/prosa.php, accessed on 1 February 2022) [34], which validated the 3D structure of proteins on the Z-score value; any fluctuation in Z-score predicted the errors in structure.

### 2.10. Docking and Interaction Analysis of MEVC

Molecular docking of vaccine constructs and human ligand TLR-3 (2A0Z), retrieved from RCSB PDB, was performed using protein–protein docking server Cluspro 2.0 (https://cluspro.bu.edu/login.php, accessed on 4 February 2022) [35]. To interpret binding interactions and binding energy evaluation, HawkDock server (http://cadd.zju.edu.cn/hawkdock/, accessed on 4 February 2022) [36] was run for the hybrid docking along with mm-GBSA analysis.

### 2.11. Codon Optimization and Computational Cloning of Vaccine Construct

The JCat tool (http://www.jcat.de/Literature.jsp, accessed on 6 February 2022) [37] was performed for the codon optimization of the vaccine construct. Initially, EMBOSS Backtranseq was run for reverse translation, and JCat optimized the gene expression with Codon Adaptation Index (CAI) > 0.8 and GC content from 40–70% in *E. coli* as reference host. Moreover, the optimized sequence was run in SnapGene software (https://www.snapgene.com/, accessed on 6 February 2022) for restriction site generation; DNA fragments and two restriction sites were inserted as desired vector for cloned plasmid formation [33].

### 2.12. Immune Simulation

C-ImmSim, a web server ImmSim (https://www.iac.rm.cnr.it/~filippo/c-immsim/index.html, accessed on 7 February 2022), was utilized to evaluate immune response generated in the form of antibodies against desired constructed antigens [38]. The immune simulation aimed to ensure the feasibility of the vaccine created with the host and assess antibodies, interferon, and cytokines generated during antigen–host interaction [39].

### 2.13. Molecular Dynamics Simulation

To ensure the stability of the vaccine construct-receptor (MEVC-TLR-3), the Imod server (https://imods.iqfr.csic.es/, accessed on 8 February 2022) was used for the molecular dynamic simulation analysis to describe the physical properties of atoms and molecules of the vaccine concerning its stability [40].

## 3. Results

### 3.1. Proteome Subtraction, Subcellular Localization, and Trans-Membrane Alpha-Helices Identification

The total proteome of *Citrobacter freundii,* containing 5119 protein sequences, was retrieved and analyzed to select putative proteins suitable for vaccine design. About 222 extracellular, outer membrane, periplasmic, and cytoplasmic localized pathogenic protein sequences were selected from the proteome [41]. Moreover, overlapping proteins representing 0.8 and 80% as cut-off value and identity percentage were excluded using the Cluster Database with High Tolerance (CD-HIT) advanced clustering algorithms. The following immunoinformatics approach was to predict the allergenicity and antigenicity of the shortlisted proteins for vaccine design. Antigenicity of shortlisted proteins was predicted by utilizing Vaxijen with a threshold value of approximately ≥0.5, and AllerTop was used to predict the allergenicity of pathogenic proteins. Approximately six proteins with high values of antigenicity, immunogenicity, nontoxicity, and nonallergen characteristics were selected for further processing and designing of the multi-epitope vaccine (MEVC). Out of six shortlisted proteins, only two proteins contained each transmembrane helix.

### 3.2. Immune Cells Epitope Prediction and Validation

Putative epitopes, B-cell epitopes, cytotoxic T-cell epitopes, and helper T-cell epitopes were identified and predicted using the Immune Epitope Database (IEDB) to design MEVC as shown in Table 1. Linear antibody or B-cell epitopes from six different protein sequences were predicted using the IEDB B-cell epitope prediction tool. Similarly, T-cell epitopes were predicted by the IEDB T-cell epitope prediction tool, which predicts the T-cell epitopes based on the binding of peptides with MHC molecules. Cytotoxic T-cell epitopes were predicted by utilizing ANN 4.0 peptide binding to the MHC-I class molecules, and peptides were sorted by their predicted score. Helper T-cells were predicted using the NN-align 2.3 (NetMHCII 2.3) method of peptide binding with MHCII and 7-allele HLA of humans as the reference set. A group of 11 B-cell epitopes, 11 CTL epitopes, and 11 HTL epitopes were selected after predicting their antigenicity and allergenicity for the *in-silico* design of MEVC, as shown in Table 2. Population coverage of MHCI and MHCII was predicted by using IEDB population coverage analysis. The population coverage of MHC-I was 90.6%, and MHC-II was 70.2%, under the threshold value of 60%, and the cumulative value of both MHC-I and MHC-II was 97.22% (Table 1 and Table 2).

### 3.3. Construction of mRNA- and Peptide-Based Vaccine

For the construction of an mRNA-based vaccine against emerging MDR *Citrobacter freundii,* highly antigenic and nonallergen epitopes were selected. In the mRNA-based vaccine construct, two epitopes of B-cell from five different protein sequences were selected, making up a total of 10 epitopes of each immune cell; and one epitope of B-cell was selected from a single protein sequence, thus making up a total of 11 B-cell epitopes. Likewise, there were 11 HTL epitopes and 11 TCL epitopes from six different protein sequences. These epitopes were joined through linkers (EAAAK, AAY, PMGLP, and GGGGS) with an additional signal peptide, five m7G Cap, Kozak sequence, and five untranslated regions (UTR) along with the addition of stop codon followed by poly-A tail for the construction of the mRNA vaccine as described in Figure 2.

Similar to the case with designing a peptide-based vaccine against emerging MDR *Citrobacter freundii,* 11 HTL, 11 B-cell, and 11 TCL epitopes from six different protein sequences were selected and linked through linkers along with adjuvants, and constructed a peptide-based MEVC with a length of 414 amino acids analyzed for antigenicity, toxicity, allergenicity, solubility, and physiochemical properties, as shown in Table 3. Interpretations showed that MEVC was nonallergen and nontoxic, with an antigenicity value of 0.9616 and solubility of 0.536 (Table 3).

### 3.4. Structure Predictions, Validation, and Docking 

The tertiary structure of the vaccine construct from Rosetta was visualized in PyMOL, as shown in Figure 3A. ProSA-web and PROCHECK, two web servers, further validated the vaccine construct, proSA-web with a Z-score value of about −7.93, as shown in Figure 3B, and ERRAT with an 80.75 quality factor. Furthermore, PROCHECK showed results in a Ramachandran plot revealing that 86.9% of the residues are present in the most favored region, 9.5% in the allowed region, and 1.1% in the allowed and 2.5% in disallowed regions, as illustrated in Figure 3C. The interaction of MEVC was evaluated by docking the 3D vaccine construct with the human toll-like receptor 3 ligand-binding domain. Cluster 2.0 was utilized to dock the TLR-3 receptor and MEVC ligand. Out of 10 predicted results, the best-docked result was selected with the lowest acquired energy score of −4755 and lowest binding energy value of −21.46 (kcal/mol) using the Hawkdock server, as illustrated in Figure 4 (Figure 3 and Figure 4).

### 3.5. Computational Cloning and Immune Simulation of Vaccine Construct

Computational cloning was performed through the codon optimization tool, JCat, which improved the DNA sequence of the vaccine construct with the enhanced value of CAI 0.96 and GC content of about 51.3. The optimized vaccine constructs, due to enhanced codon sequence, provided better results when subjected to protein expression in *E. coli,* and then formed DNA fragments exploited in the pET28a-MH6 vector for cloning, as illustrated in Figure 5. C-ImmSim is an immune simulation server that evaluates vaccine interaction with the host as an antigen and interprets a robust immune response in antibody production, as shown in Figure 6. The result showed that the vaccine as an antigen produces a strong response in antibody production, as represented in Figure 5. The antibody production was more than 700,000 in 2 to 3 days, but the response was neutralized entirely on the fifth day. Specific antibodies such as IgM, IgG1, and IgG2 were noticed to be increased during the same period. Results showed that the vaccine produced a positive immune response (Figure 5 and Figure 6).

### 3.6. Molecular Dynamics Simulations

Molecular dynamics simulations interpret results in B-factor mobility, and atomic index graph predicts the vaccine construct reliability in the environment. The simulation uses the I-Mods web server, as shown in Figure 7. B factor, stability, flexibility, residues covariance map, and atom index of the complex were checked and interpreted.

## 4. Discussion

*Citrobacter freundii* is the most infectious species of *Citrobacter* spp., which efficiently play a role in an array of nosocomial and community-acquired infections [42]. Inherent and acquired characteristics of *Citrobacter freundii* impede the activity of broad-spectrum antibiotics, thereby emerging as highly opportunistic multidrug-resistant *Citrobacter freundii.* MDR *Citrobacter freundii* has become challenging in regard to treating its infections, so there is a need for a new immunoinformatics-approach-based therapy to treat such infections [4]. This approach is considered adequate as it boosts the infected person’s immunity, which protects the person for their whole life [43,44].

This study aimed to construct a vaccine against the emerging multidrug-resistant *Citrobacter freundii* to treat its infections effectively. A whole proteome was retrieved, and pathogenic nonhomologous proteins were screened. These screened proteins were further analyzed to exclude overlapping proteins, and transmembrane helices analysis was performed to predict signal transduction across the cell membrane [45]. Immune cell epitopes were also selected from the six shortlisted protein sequences based on their antigenicity and allergenicity activity. An epitope with antigenicity above a threshold value of 0.5 and nonallergen activity was desired. In this way, 11 each of B-cell, TCL, and HTL epitopes were selected as suitable targets for vaccine design [46]. Multi-epitope peptide- and mRNA-based vaccines were constructed through the assistance of linkers EAAK, AAY, GPGPG, and KK, and adjuvants in the case of the peptide-based vaccine only. Multi-epitope peptide- and mRNA-based vaccines constitute 414 amino acids and 692 with 0.9 and 1 antigenicity scores. These vaccine constructs demonstrate nonallergenic, nontoxic, and solubility scores of 0.53 and 0.76, respectively. Secondary and tertiary structures of vaccines were predicted utilizing different web servers. The tertiary structure was refined for a stable system and enhanced characteristics. Previously, the immunoinformatics-approach-based vaccine designs were used to design the vaccine for toxoplasmosis [44,47,48].

Molecular docking analyses were performed between the vaccine construct human toll-like receptor 2A0Z docked results, with the lowest binding affinity selected as it shows a bond strength between receptor and vaccine. Further, validations of vaccine construct were carried out to predict the best interpretations, stability, and immune-level interaction following the protocol from a previous study [49]. Initially, a positive immune response upon antigen introduction in the host was predicted using IgM, IgG1, and IgG2 antibodies. These interpretations indicated that the vaccine is proficient in generating diverse and healthier immune responses [50].

The primary objective of creating a vaccination was to induce long-term memory. B- and T-cell activation must coincide with accomplishing this production. The host will be prepared to produce a rapid and efficient defense against future infections. However, the vaccine’s success relies heavily on strategically deploying a subset of antigens called epitopes. As a result, it is crucial to locate the epitope that may trigger the merging of both cells to develop an effective vaccination. In addition to IL-4 and IFN-, the HTL can also produce IL-10. Human T-lymphotropic (HTL) epitopes are shown on antigen-presenting cells (APCs), and lymphocytes can release chemokines that play critical roles in the immune system’s defense against these microorganisms. After successfully eliminating infection, all immune cells, except memory cells, perish. Recognizing the antigenic epitopes is accomplished by the B-cell membrane-bound immunoglobin receptor. This way, the epitopes are induced and processed before being presented to T-cells via MHC class-II molecules. T-cell receptors (TCR) are specific for recognizing them. Because of this, B-cells undergo a differentiation process into plasma cells, which then generate antibodies to fight off invading memory and foreign cells. Vaccines effectively eradicate infections or diseases for the long term with minimal chances of resistance compared to other therapeutic measures such as antibiotics. The vaccine can produce enduring immunity in the host through acquired immunity. Against *Citrobacter freundii*, this vaccine could be a potential therapeutic candidate to combat various nosocomial and community-acquired infections.

## 5. Conclusions

A potential therapeutic strategy is needed to restrict the MDR *Citrobacter freundii*. Therefore, an immunoinformatics-approach-based mRNA vaccine was designed from selected putative epitopes. Physiochemical properties evaluation and immune simulation demonstrated the potency of the vaccine construct. The vaccine design elicits the anticipated physiochemical and immune responses. The immunization was shown to produce an immune response that was consistent with our goals in the immunological simulation. This framework should be considered as a potential candidate for in vitro and in vivo research in contraindications to *Citrobacter freundii*, with numerous serological tests performed to validate the trigger of reaction on demand.

## Figures and Tables

**Figure 1 medicina-58-01356-f001:**
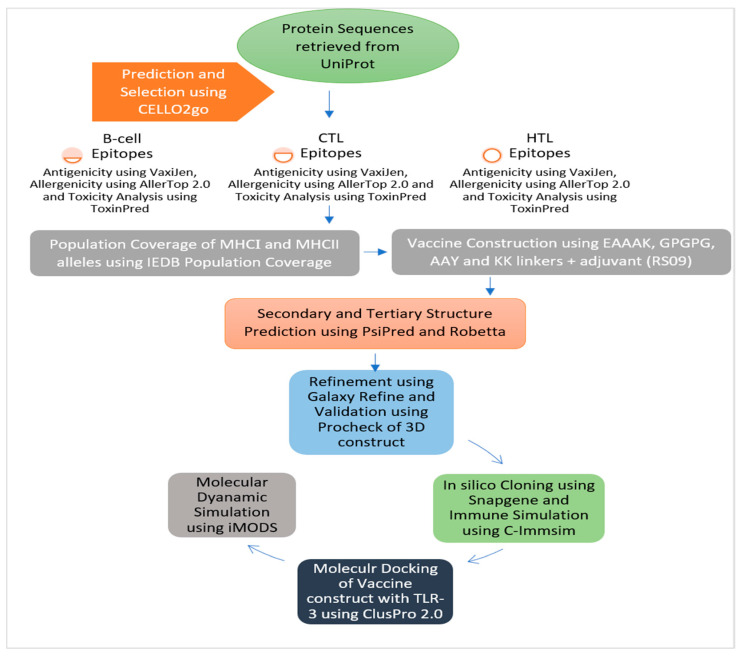
Flowchart diagram of designing mRNA- and peptide-based vaccine construct against emerging MDR *Citrobacter freundii*.

**Figure 2 medicina-58-01356-f002:**
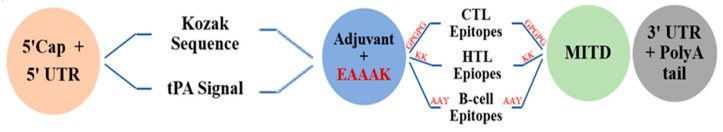
Pictorial description of vaccine construct.

**Figure 3 medicina-58-01356-f003:**
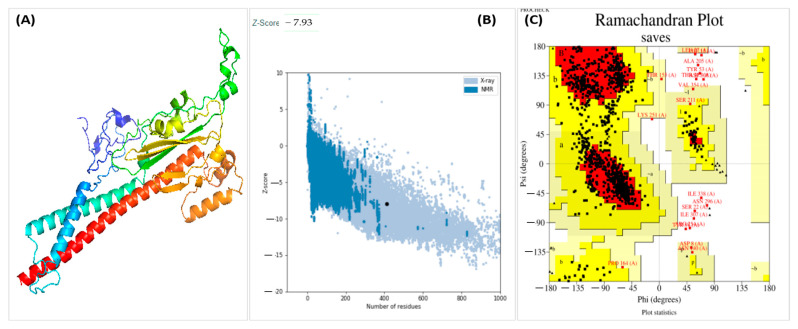
(**A**) Predicted 3D structure of vaccine construct. (**B**) ProSA web with *z*-score indicates overall model quality and recognize errors in 3D model of vaccine construct. (**C**) Ramachandran plot describes the validation of the vaccine construct by showing amino acids in favored and allowed regions.

**Figure 4 medicina-58-01356-f004:**
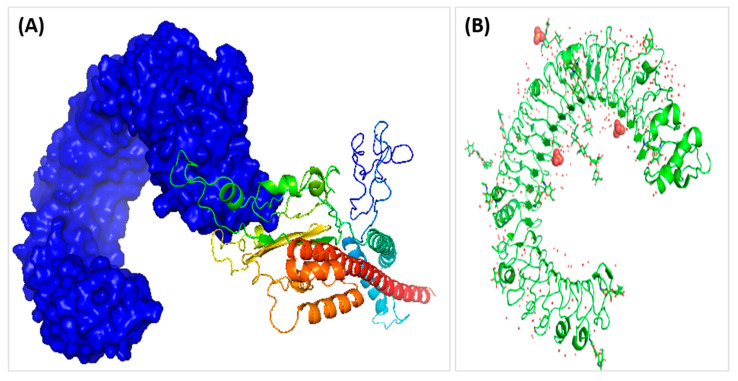
(**A**) 3D docked structure of vaccine constructs and TLR-3 using Cluspro 2.0 in which the construct acts as receptor and TLR_3 as the legend for the formation of docked complex. (**B**) 2A0Z is used as the legend for docking with vaccine construct.

**Figure 5 medicina-58-01356-f005:**
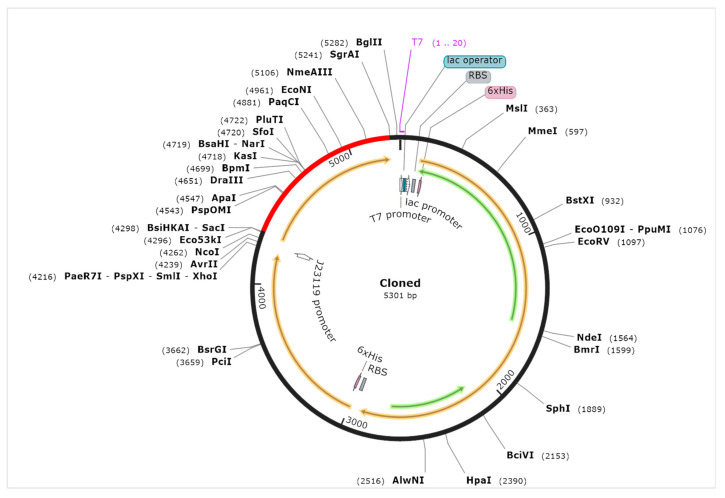
Codon optimization and computational cloning of vaccine construct using JCat and SnapGene tool.

**Figure 6 medicina-58-01356-f006:**
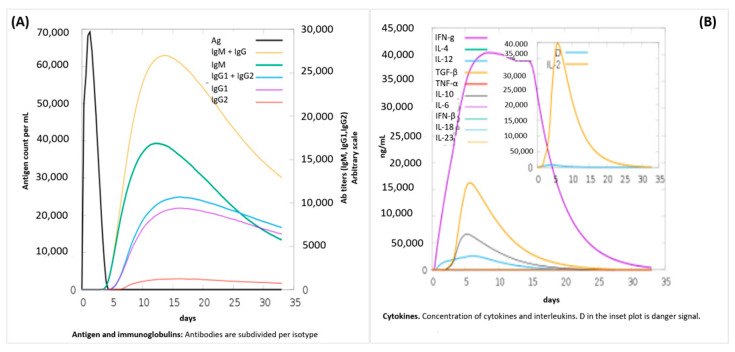
In-silico simulation of immune response using the vaccine as an antigen to generate a robust immune response. (**A**) Representation of antigen and immunoglobulins in immune response. (**B**) Representation of cytokines in immune response.

**Figure 7 medicina-58-01356-f007:**
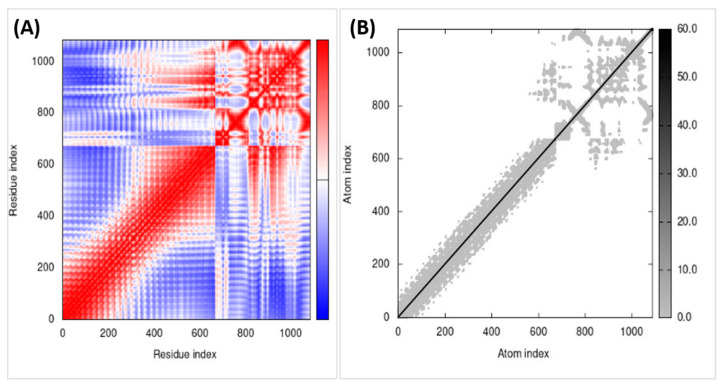
(**A**) Covariance matrix indicates coupling between pairs of residues, i.e., whether they experience correlated (red), uncorrelated (white), or anticorrelated (blue) motions. (**B**) The elastic network model defines which pairs of atoms are connected by springs. Each dot in the graph represents one spring between the corresponding pair of atoms. Dots are colored according to their stiffness; the darker grays indicate stiffer springs, and vice versa.

**Table 1 medicina-58-01356-t001:** Population coverage analysis of the selected MHC-I and MHC-II epitopes of the vaccine construct, as predicted by the IEDB server.

Population/Area	Class Combined		
	Coverage	Average Hit	pc90
World	97.22%	4.27	1.71
Average	97.22%	4.27	1.71
Standard Deviation	0	0	0

**Table 2 medicina-58-01356-t002:** Predicted B-cell lymphocyte (BCL), cytotoxic T-cell lymphocyte (CTL), and helper T-cell lymphocyte (HTL) epitopes with their antigenicity and allergenicity.

Target Protein Accession ID	Type of Epitopes	Total Shortlisted Epitopes	Epitope Sequences	TMHMM Score	Antigenicity Score
>UPI000021C373	BCL	2	ISDKKFYNLSDTSGKGSLDYPLP	0	0.9
			PARLNPVSGKVSPH		1.2
	CTL	1	RLNPVSGKV		1.42
	HTL	2	LHYELVINNNPVNSL		0.82
			HLHYELVINNNPVNS		1.01
>UPI000272882E	BCL	2	SGSKSSDTGSYSG	0	1.8
			GSSDATT		2.7
	CTL	2	GSKSSDTGSY		1.7
			FQIRYRATAI		1.06
	HTL	1	NKGIDISAARGTPIY		0.8
>UPI0002B869E4	BCL	1	VSINDLKQWNNLRGSSLK	0	0.7
	CTL	2	YLHDVTLRA		1.04
			PTDFWSLPL		1.18
	HTL	2	GRVMKAIKTNKARGK		0.57
			PQYVMVPKKHAEKLR		0.62
>UPI0004D735E6	BCL	2	FAGENGNSTT	0	1.6
			SGTDYGRS		0.9
	CTL	2	ALKNVPGVGA		1.12
			APSAGTSAL		1.06
	HTL	2	KLGYRVNRNLDFQLN		1.6
			DINQVIGDTTAARLN		0.8
>UPI0004DAE55E	BCL	2	ADERDQLKSIQADIAAKERAVRQQQQQRSTLLA	1	0.67
			SSGGQGR		4
	CTL	2	FSVRPLFYA		1.16
			ASRKGSTYK		1.73
	HTL	2	RDQLKSIQADIAAKE		0.64
			ERDQLKSIQADIAAK		0.7
>UPI000C150917	BCL	2	DQTDSVAVSH	1	0.8
			GNSTSGQRGNN		2.9
	CTL	2	ASRKGSTYK		1.73
			GEQLQGELRW		1.17
	HTL	2	DFSFRLNGNLDKTQA		1.69
			GDFSFRLNGNLDKTQ		1.44

The antigenicity status of all predicted epitopes was antigenic, while the allergenicity was nonallergen.

**Table 3 medicina-58-01356-t003:** Vaccine constructs and their physiochemical properties are predicted using ProtParam, VaxiJen, AllerTop, ToxinPred, and SOLpro.

**Vaccine Type**	Proteome-Wide Peptides Based on MEVC Construct	Number of Amino Acids	Molecular Weight (kd)	Theoretical pI	Aliphatic Index	Hydropathicity (GRAVY)	Antigenicity Score	Solubility
Peptide	EAAAKISDKKFYNLSDTSGKGSLDYPLPPARLNPVSGKVSPHSGSKSSDTGSYSGGSSDATTVSINDLKQWNNLRRLANNSDSFAGENGNSTTSGTDYGRSADERDQLKSIQADIAAKERAVRQQQQQRSTLLASSGGQGRDQTDSVAVSHGNSTSGQRGNNCPGPGRLNPVSGKVGSKSSDTGSYFQIRYRATAIYLHDVTLRAPTDFWSLPLALKNVPGVGAAPSAGTSALFSVRPLFYAASRKGSTYKASRKGSTYKGEQLQGELRWAAYHLHYELVINNNPVNSLHYELVINNNPVNSLNKGIDISAARGTPIYGRVMKAIKTNKARGKPQYVMVPKKHAEKLRKLGYRVNRNLDFQLNDINQVIGDTTAARLNRDQLKSIQADIAAKEERDQLKSIQADIAAKHHHHHH	414	44,938.96	9.86	71.47	−0.731	0.9616	0.536
mRNA	EAAAKMKNARTTLIAAAIAGTLVTTSPAGIANADDAGLDPNAAAGPDAVGFDPNLPPAPDAAPVDTPPAPEDAGFDPNLPPPLAPDFLSPPAEEAPPVPVAYSVNWDAIAQCESGGNWSINTGNGYYGGLRFTAGTWRANGGSGSAANASREEQIRVAENVLRSQGIRAWPVCGRRGGPGPGLHYELVINNNPVNSLGPGPGHLHYELVINNNPVNSGPGPGNKGIDISAARGTPIYGPGPGGRVMKAIKTNKARGKGPGPGPQYVMVPKKHAEKLRGPGPGKLGYRVNRNLDFQLNGPGPGDINQVIGDTTAARLNGPGPGRDQLKSIQADIAAKEGPGPGERDQLKSIQADIAAKGPGPGDFSFRLNGNLDKTQAGPGPGGDFSFRLNGNLDKTQKKISDKKFYNLSDTSGKGSLDYPLPKKPARLNPVSGKVSPHKKSGSKSSDTGSYSGKKGSSDATTKKVSINDLKQWNNLRGSSLKKKGSSAQRLANNSDSKKFAGENGNSTTKKSGTDYGRSKKSSGGQGRKKDQTDSVAVSHKKGNSTSGQRGNNAAYRLNPVSGKVAAYGSKSSDTGSYAAYFQIRYRATAIAAYYLHDVTLRAAAYPTDFWSLPLAAYALKNVPGVGAAAYAPSAGTSALAAYFSVRPLFYAAAYASRKGSTYKAAYASRKGSTYKAAYGEQLQGELRWAAY	692	71,897.13	9.8	62.2	−0.63	1.08	0.764

The antigenicity status of all predicted epitopes was antigenic, allergenicity was nonallergen, and the toxicity was nontoxic.

## Data Availability

More data related to this study can be accesses upon a reasonable request to the corresponding authors.

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
