# Peer review of "Designing mRNA- and Peptide-Based Vaccine Construct against Emerging Multidrug-Resistant Citrobacter freundii: A Computational-Based Subtractive Proteomics Approach"

_medicina, 2022, doi:10.3390/medicina58101356_

Round 1

Reviewer 1 Report

The authors designed mRNA, and peptide-based vaccine constructs against Citrobacter freundii using proteomics data and in silico prediction tools.

While the authors used prediction tools to develop the vaccine constructs, I think their methodology does not bring any novelty to the table.  I do not think the paper advances the current knowledge. I think they need to make these constructs and test whether they are binding to C. freundii infected animals or human serum. They used in silico tools to predict the immune response, but I do not think it's the appropriate way to test whether their constructs are immunogenic or not. They can perform in vivo animal vaccination studies to observe that. You need to validate your results. I think the paper is not ready for publication at the current stage. I think they need to add some in vitro/in vivo data.  In addition, I think the manuscript needs extensive English editing.

Can you define clearly what they used as an adjuvant in the peptide construct? Is it only β-defensin? Can you illustrate your peptide constructs as a figure? 

What does Figure 1B, and C illustrates? Please explain it in the results.
Figure 2: Can you add to the figure legends the 3D structures you used for TLR3? 
What is the reasoning behind using only the TLR3 structure for docking in Figure 2? Why you didn't use TLR2/4 or 5, please explain. 

Line 145 -  β-defesin --> β-defensin

What is the purpose of Figure 5? How did you interpret it?

Bacteria do not have gender. It is not "he" or "she." Please change it accordingly in line 294. 

Author Response

Reviewer 1

Comments and Suggestions for Authors

The authors designed mRNA, and peptide-based vaccine constructs against Citrobacter freundii using proteomics data and in silico prediction tools.

While the authors used prediction tools to develop the vaccine constructs, I think their methodology does not bring any novelty to the table.  I do not think the paper advances the current knowledge. I think they need to make these constructs and test whether they are binding to C. freundii infected animals or human serumThey used in silico tools to predict the immune response, but I do not think it's the appropriate way to test whether their constructs are immunogenic or not. They can perform in vivo animal vaccination studies to observe that. You need to validate your results. I think the paper is not ready for publication at the current stage. I think they need to add some in vitro/in vivo data.  In addition, I think the manuscript needs extensive English editing.

Response: Dear reviewer, we would like to acknowledge your kind efforts for the current manuscript. We must appreciate that after addressing the comments from you and other reviewers, the current manuscript become more suitable for the reader and scientific community.

While addressing your comment about the in vitro/in vivo analysis, we would like to explain that we have first constructed an in-silico based vaccine and done its immune simulation for confirmation of robust immune response and molecular dynamic simulations after the formation of docked complex with TLR-3 for its stability. This study was purely computational based and we followed the protocol using previously published studies in well-known journals for example (Khalid et al., 2022), (Basak et al., 2021), (Shey et al., 2019), (Naveed et al., 2022). All of them performed computational based analysis. And yes, we understand that the in vitro/in vivo analysis will enhance the quality of study but unfortunately at the current stage we are unable to perform the experiments as we are lacking of funding to conduct a wet lab-based study. But once we have enough funding to conduct the experiments, we will go for wet lan analysis (in vitro/in vivo analysis).

Basak, S., D. Deb, U. Narsaria, T. Kar, F. Castiglione, I. Sanyal, P.D. Bade, and A.P. Srivastava. 2021. In silico designing of vaccine candidate against Clostridium difficile. Scientific Reports. 11:14215.

Khalid, K., S. Irum, S.R. Ullah, and S. Andleeb. 2022. In-Silico Vaccine Design Based on a Novel Vaccine Candidate Against Infections Caused by Acinetobacter baumannii. Int J Pept Res Ther. 28:16.

Naveed, M., K. Jabeen, R. Naz, M.S. Mughal, A.A. Rabaan, M.A. Bakhrebah, F.M. Alhoshani, M. Aljeldah, B.R.A. Shammari, and M. Alissa. 2022. Regulation of Host Immune Response against Enterobacter cloacae Proteins via Computational mRNA Vaccine Design through Transcriptional Modification. Microorganisms. 10:1621.

Shey, R.A., S.M. Ghogomu, K.K. Esoh, N.D. Nebangwa, C.M. Shintouo, N.F. Nongley, B.F. Asa, F.N. Ngale, L. Vanhamme, and J. Souopgui. 2019. In-silico design of a multi-epitope vaccine candidate against onchocerciasis and related filarial diseases. Scientific Reports. 9:4409.

Apart from this, we have thoroughly revised the manuscript for English proofreading and grammatical mistakes.

Can you define clearly what they used as an adjuvant in the peptide construct? Is it only β-defensin? Can you illustrate your peptide constructs as a figure? 

Response: Dear reviewer, yes, the β-defensin was used as an adjuvant in the peptide construct. Furthermore, we have added illustrated our peptide constructs as a figure in the revised manuscript.

What does Figure 1B, and C illustrates? Please explain it in the results.
Figure 2: Can you add to the figure legends the 3D structures you used for TLR3? 
What is the reasoning behind using only the TLR3 structure for docking in Figure 2? Why you didn't use TLR2/4 or 5, please explain. 

Response: Dear reviewer, In Fig. 4B, z-score indicates overall model quality and recognize errors in 3D model of vaccine construct. In Fig. 4C, Ramachandran plot describe the validation of vaccine construct by showing amino acids in favoured and allowed regions.

The reasoning behind using only the TLR3 structure for docking was that it gives best values of energy interaction of the vaccine construct and TLR-3 docked complex.

Line 145 - β-defesin --> β-defensin

Response: Corrected as β-defensin

What is the purpose of Figure 5? How did you interpret it?

Response: In Fig. 8A) Covariance matrix indicates coupling between pairs of residues, i.e., whether they experience correlated (red), uncorrelated (white) or anti-correlated (blue) motions.

In Fig. 8B) The elastic network model defines which pairs of atoms are connected by springs. Each dot in the graph represents one spring between the corresponding pair of atoms. Dots are colored according to their stiffness, the darker grays indicate stiffer springs and vice versa.

Bacteria do not have gender. It is not "he" or "she." Please change it accordingly in line 294.

Response: Line 311: The sentence has been corrected.

Reviewer 2 Report

The author explored the ability of bioinformatics tools in designing a mRNA / peptide based vaccine against MDR Citrobacter freundii. This approach had been used in the development of several vaccine and immunotherapy over the recent years, which is a good approach in looking for possible antigenic-epitope candidate for the intended aim of the study. This article will provide a new knowledge and will benefit the development of such vaccine for the particular pathogen. Nonetheless, several revisions need to be addressed at it current stage, to make it a well-written manuscript

Revisions

1. Result section:

a) Table and Figure: require revision in the title and figure legend. The title need to be more specific and adequately explanatory for example Table 1: Population Coverage Analysis of what component?  Additionally figure legend/ label need to be detailed while being concise. 

b) Result text: please go through the text for spelling/ grammatical mistakes, i.e Line 199: shortlisted instead of two words with comma in between.

2. Discussion and Conclusion sections:

This sections require re-written as  this current state, it is a mere repetition of the results-without additional discussion or comment on the finding from the author. The author possibly need to comment/ provide the information on the role or biological pathway involve in the particular region of epitope selected. Additionally, the predicted immunogenicity need further explanation on the reliability based on several other studies done before and also their correlation in predicting in-vitro and in-vivo responses. 

One main thing to comment, the author need to be careful in making statement on the impact towards the actual disease/infection-as this approach only help in designing the vaccine: the actual vaccine still need top be developed and go through the whole stages of laboratory and clinical trial before it can implies it usability. (Refer to statement in Line 299-300). 

Also it is preferred to use a more scientifically sound term i.e. Referring to Citrobacter freundii as "he" (Line 294). 

Suggestions:

1. Methodology section: concisely described and satisfactory adequate. Suggestion is to include a flow diagram on the different stages and software used for clearer and easier reference.  

2. Result: A flow diagram to showcase the overview of the results at different stages of the bioinformatic analysis (i.e. how many protein sequence were retrieved, how many were selected based on criteria etc).

Thank you

Author Response

Reviewer 2

Comments and Suggestions for Authors

The author explored the ability of bioinformatics tools in designing a mRNA / peptide based vaccine against MDR Citrobacter freundii. This approach had been used in the development of several vaccine and immunotherapy over the recent years, which is a good approach in looking for possible antigenic-epitope candidate for the intended aim of the study. This article will provide a new knowledge and will benefit the development of such vaccine for the particular pathogen. Nonetheless, several revisions need to be addressed at it current stage, to make it a well-written manuscript

Response: Dear reviewer, we would like to acknowledge your kind efforts for the current manuscript. We must appreciate that after addressing the comments from you and other reviewers, the current manuscript become more suitable for the reader and scientific community. The current manuscript has been thoroughly revised as per comments from you and other reviewers.

Revisions

  1. Result section:
  2. a) Table and Figure: require revision in the title and figure legend. The title need to be more specific and adequately explanatory for example Table 1: Population Coverage Analysis of what component?  Additionally figure legend/ label need to be detailed while being concise. 

Response: Dear reviewer, the legends for figures and tables has been revised.

Table 1. Population Coverage Analysis of the selected MHCI and MHCII epitopes of the vaccine construct, as predicted by IEDB server

Table 2. Predicted B-cell, CTL and HTL Epitopes with their Antigenicity and Allergenicity.

Table 3. Vaccine constructs and their Physiochemical Properties predicted by using ProtParam, VaxiJen, AllerTop, ToxinPred and SOLpro.

Figure 4. (A): Predicted 3D Structure of vaccine construct (B) ProSA web with z-score indicates overall model quality and recognize errors in 3D model of vaccine construct. (C) Ramachandran Plot describe the validation of vaccine construct by showing amino acids in favoured and allowed regions

Figure 5. 3D Docked Structure of vaccine construct and TLR-3 using Cluspro 2.0 in which construct acting as receptor and TLR_3 as legend for the formation of docked complex

Figure 6. Codon Optimization and Computational cloning of Vaccine Construct using JCat and Snapgene tool

Figure 7. In silico simulation of immune response using vaccine as an antigen to generate robust immune response.

Figure 8. (A) Covariance matrix indicates coupling between pairs of residues, i.e. whether they experience correlated (red), uncorrelated (white) or anti-correlated (blue) motions.and (B) The elastic network model defines which pairs of atoms are connected by springs. Each dot in the graph represents one spring between the corresponding pair of atoms. Dots are colored according to their stiffness, the darker grays indicate stiffer springs and vice versa.

  1. b) Result text: please go through the text for spelling/ grammatical mistakes, i.e Line 199: shortlisted instead of two words with comma in between.

Response: Line 206: The sentence has been revised and corrected.

  1. Discussion and Conclusion sections:

This sections require re-written as this current state, it is a mere repetition of the results-without additional discussion or comment on the finding from the author. The author possibly need to comment/ provide the information on the role or biological pathway involve in the particular region of epitope selected. Additionally, the predicted immunogenicity need further explanation on the reliability based on several other studies done before and also their correlation in predicting in-vitro and in-vivo responses. 

Response: Line 342-356:

Discussion

The primary objective of creating a vaccination was to induce long-term memory. B and T cell activation must coincide with accomplishing this production. This way, the host will be prepared to mount a rapid and efficient defense against future infections. However, the vaccine's success relies heavily on strategically deploying a subset of antigens called epitopes. As a result, it is crucial to locate the epitope that may trigger the merging of both cells to develop an effective vaccination. In addition to IL-4 and IFN-, the HTL can also produce IL-10. Human T-lymphotropic (HTL) epitopes are shown on antigen-presenting cells (APCs), and lymphocytes can release chemokines that play critical roles in the immune system's defense against these microorganisms. After successfully eliminating infection, all immune cells, except memory cells, perish. Recognizing the antigenic epitopes is accomplished by the B-cell membrane-bound immunoglobin receptor. This way, the epitopes are induced and processed before being presented to T-cells via MHC class-II molecules. T-cell receptors are specific for recognizing them (TCR). Because of this, B-cells undergo a differentiation process into plasma cells, which then generate antibodies to fight off invading memory and foreign cells.

Conclusion: Line 365-370:

The vaccine design elicits the anticipated physiochemical and immune responses. The immunization was shown to produce an immune response that was consistent with our goals in the immunological simulation. This framework should be considered as a potential candidate for in vitro and in vivo research in contraindications to Citrobacter freundii, with numerous serological tests performed to validate the trigger of reaction on demand.

One main thing to comment, the author need to be careful in making statement on the impact towards the actual disease/infection-as this approach only help in designing the vaccine: the actual vaccine still need top be developed and go through the whole stages of laboratory and clinical trial before it can implies it usability. (Refer to statement in Line 299-300). 

Response: Dear reviewer, this study was based on computational based approach so statement about treatment of infections given on the basis of immune simulation results. Vaccine injection as antigen resulted in the robust production of antibodies which could help in eradicating citrobacter infection before they damage the body.

Also it is preferred to use a more scientifically sound term i.e. Referring to Citrobacter freundii as "he" (Line 294). 

Response: Line 311: The sentence has been revised and corrected.

Suggestions:

  1. Methodology section: concisely described and satisfactory adequate. Suggestion is to include a flow diagram on the different stages and software used for clearer and easier reference.  
  2. Result: A flow diagram to showcase the overview of the results at different stages of the bioinformatic analysis (i.e. how many protein sequence were retrieved, how many were selected based on criteria etc).

Response: Dear reviewer, thank you for your valuable suggestion. We have added figure 1 and 2 in the revised version of manuscript.

Reviewer 3 Report

This study uses modern computational-based vaccine design approach (also known as reverse vaccinology) to develop a vaccine against the opportunistic MDR C. freundii. This in silico design permits the optimal use of resources for effective vaccine development. It is a much needed study which can assist in reducing the time/resources needed for the development of an efficacious and effective vaccine to combat MDR pathogens. The manuscript is well written, however, addressing the following points will make the message clearer and the experiments easier to repeat;

1  1.  Immune stimulation and immune simulation are being used interchangeably. Kindly check and rectify.

22. Section 2.6, page 3; what criteria were used to ascertain a putative peptide as “best” for the design of an mRNA-based vaccine against C. freundii?

  3.  Were these linkers (EAAK, AAY, 147 GPGPG, and SKK) used to link the 11 peptide of each epitope type (B-cell, cytotoxic T-cell, and Helper T-cell epitopes) or they were used to link all 33 peptides into a single construct?

44.  Why and how was the six proteins selected?

55.  Out of six shortlisted proteins, only two proteins contained each transmembrane helix. But it is known that proteins with a transmembrane domain and/or signal peptide are easily exposed to the immune system.  So, why were proteins with no transmembrane domain included in the study?

66.  Section 3.2: The content of Table 1 and 2 is different from their description in the text.

77.  Section 3.2: Please give the full meaning of BCL.

88.  Section 3.2: The text mentions that a group of 11 B cell epitopes were selected but Table 2 shows 12 B cell epitopes. Section 3.3 also mentions 11 B cell epitopes.

99. Section 3.5, line 276; which figure is being referred to?

Author Response

Reviewer 3

Comments and Suggestions for Authors

This study uses modern computational-based vaccine design approach (also known as reverse vaccinology) to develop a vaccine against the opportunistic MDR C. freundii. This in silico design permits the optimal use of resources for effective vaccine development. It is a much needed study which can assist in reducing the time/resources needed for the development of an efficacious and effective vaccine to combat MDR pathogens. The manuscript is well written, however, addressing the following points will make the message clearer and the experiments easier to repeat;

  1. Immune stimulation and immune simulation are being used interchangeably. Kindly check and rectify.

Response: Line 94: Immune simulation are being used and rectify.

  1. Section 2.6, page 3; what criteria were used to ascertain a putative peptide as “best” for the design of an mRNA-based vaccine against C. freundii?

Response: Dear reviewer, Antigenic, Non-allergenic, and non-toxic epitopes were selected using VaxiJen, AllerTOP 2.0 and ToxinPred for the construction of putative peptides with addition of linkers like (EAAAK, AAY, PMGLP, and GGGGS) and adjuvant β-defensin to improve the immunogenicity of the construct.”

  1. Were these linkers (EAAK, AAY, GPGPG, and KK) used to link the 11 peptide of each epitope type (B-cell, cytotoxic T-cell, and Helper T-cell epitopes) or they were used to link all 33 peptides into a single construct?

Response: One of the linkers is KK not SKK and it is rectified. These linkers (EAAK, AAY, GPGPG, and KK) used to link the 11 peptide of each epitope type (B-cell, cytotoxic T-cell, and Helper T-cell epitopes) as depicted in the Table 3 of vaccine construct.

  1. Why and how was the six proteins selected?

Response: Six proteins were selected from a whole proteome of Citrobacter freundii retrieved from UniProt. Pathogenic and extracellular membrane were shortlisted using CELLO2GO web server. Human host homologous proteins were omitted using BLASTp analysis and shortlisted the non-homologous proteins for vaccine targets. Furthermore, on the basis of antigenicity, allergenicity and toxicity six putative epitopes were selected for the construction of mRNA vaccine construct.

  1. Out of six shortlisted proteins, only two proteins contained each transmembrane helix. But it is known that proteins with a transmembrane domain and/or signal peptide are easily exposed to the immune system.  So, why were proteins with no transmembrane domain included in the study?

Response: Those four proteins are present in extracellular region of the membrane and they also exposed to the immune system for robust response.

  1. Section 3.2: The content of Table 1 and 2 is different from their description in the text.

Response: The population coverage of MHCI was 90.6% and MHCII was 70.2%, under the threshold value of 60% and the cumulative value of both MHCI and MHCII became 97.22% as illustrated in the table 1.

 The description of Table 2 is right there was a mistake in Table 2 and it is rectified. Total 11 B cell epitopes were selected for the vaccine construct.

  1. Section 3.2: Please give the full meaning of BCL.

Response: Line 231-232: The abbreviation has been written.

  1. Section 3.2: The text mentions that a group of 11 B cell epitopes were selected but Table 2 shows 12 B cell epitopes. Section 3.3 also mentions 11 B cell epitopes.

Response: The description of Table 2 is right there was a mistake in Table 2 and it is rectified. Total 11 B cell epitopes were selected for the vaccine construct.

  1. Section 3.5, line 276; which figure is being referred to?

Response: C-Immsim is an immune simulation server that evaluates vaccine interaction with the host as an antigen and interprets a robust immune response in antibody production, as shown in figure 7.

Round 2

Reviewer 2 Report

Thank you so much for the revisions provided. It is a better and clearer read in its current state compared to previous version. 

One main thing that require re-look/revision is Figure 2, which I believe are missing words in some boxes. Additionally, other than repeating it again from Figure 1, the main aim of Figure 2 is to summarize the flow that had been done-so it can make it more concise. And this need to be cited to its respective text, i.e: Methodology and Result section where it will be relevant. 

Discussion and conclusion had improved significantly, although more points can be added, yet it is already adequately written in the current version. Thank you

Author Response

Reviewer 2

Comments and Suggestions for Authors

Thank you so much for the revisions provided. It is a better and clearer read in its current state compared to previous version. One main thing that require re-look/revision is Figure 2, which I believe are missing words in some boxes. Additionally, other than repeating it again from Figure 1, the main aim of Figure 2 is to summarize the flow that had been done-so it can make it more concise. And this need to be cited to its respective text, i.e: Methodology and Result section where it will be relevant. 

Response: Dear reviewer, we would like to say our sincere thanks to you for your kind efforts to our manuscript. We must appreciate that the manuscript becomes better because of comments from you and other reviewers. Furthermore, your current comment from figure 1 and 2 has been addressed. The figure 2 has been merged with figure 1 and figure 1 also has been cited in the material and method section of the revised manuscript.

Discussion and conclusion had improved significantly, although more points can be added, yet it is already adequately written in the current version. Thank you

Response: Thank you dear reviewer for your valuable comment that the discussion is adequately written. This is definitely after addressing your last comments. Furthermore, we have added a small part of discussion in the revised manuscript with more references to strengthen the statements.